# Polarization Angle Dependence of Optical Gain in a Hybrid Structure of Alexa-Flour 488/M13 Bacteriophage

**DOI:** 10.3390/nano11123309

**Published:** 2021-12-06

**Authors:** Inhong Kim, Juyeong Jang, Seunghwan Lee, Won-Geun Kim, Jin-Woo Oh, Irène Wang, Jean-Claude Vial, Kwangseuk Kyhm

**Affiliations:** 1Department of Opto/Cogno-Mechatronics Engineering, Research Center for Dielectric Advanced Matter Physic (RCDAMP), Pusan National University, Busan 46241, Korea; enho@pusan.ac.kr (I.K.); quantum375@gmail.com (J.J.); seunghwanlee00@gmail.com (S.L.); 2Department of Nano Fusion Technology, Pusan National University, Busan 46241, Korea; kim1guen@gmail.com (W.-G.K.); ojw@pusan.ac.kr (J.-W.O.); 3Laboratoire Interdisciplinaire de Physique (LiPhy), Centre National de la Recherche Scientifique (CNRS), Université Grenoble Alpes, 38000 Grenoble, France; irene.wang@univ-grenoble-alpes.fr (I.W.); jeanclaudevial@gmail.com (J.-C.V.)

**Keywords:** bio-laser, optical gain, amplified spontaneous emission, M13 bacteriophage

## Abstract

We measured optical modal gain of a dye–virus hybrid structure using a variable stripe length method, where Alexa-fluor-488 dye was coated on a virus assembly of M13 bacteriophage. Inspired by the structural periodicity of the wrinkle-like virus assembly, the edge emission of amplified spontaneous emission was measured for increasing excited optical stripe length, which was aligned to be either parallel or perpendicular to the wrinkle alignment. We found that the edge emission showed a strong optical anisotropy, and a spectral etalon also appeared in the gain spectrum. These results can be attributed to the corrugated structure, which causes a similar effect to a DFB laser, and we also estimated effective cavity lengths.

## 1. Introduction

Currently, bio-lasers are attracting a great deal of attention for biomedical and biological applications, whereby micro-cavities are integrated with biochemical materials [1,2,3]. The lasing of bio-lasers is often evaluated in terms of the narrowness of their spectral linewidth, the intensity of their super-linearity for excitation, and their Q-factor. However, optical gain is also an important factor with respect to the efficiency of lasing action in micro-bio-cavities [4,5,6]. To date, optical gain in bio-lasers has not drawn much attention. This can mainly be attributed to biocompatibility and chemical stability, which limit the diversity of bio-laser gain medium, but enhanced mode confinement and degree polarization can be achieved, provided that the gain medium possesses a self-assembled structure.

For electronic and optical device applications, self-assembly methods are widely used due to their simple and cost-effective processes [7]. Among the various biochemical self-assembled structures, the M13 bacteriophage has great potential for optical imaging and biosensor applications [8,9,10,11,12,13] due to its site-specific binding properties and well-defined cylindrical shape (880 nm length and 6.6 nm diameter) [9]. Using the dip-coating method, a periodic structure can be assembled with M13 bacteriophage [13,14,15]. As the period can be modified with targeted molecules bound to the M13 bacteriophage, the corrugated structure is useful for optical grating. For example, a specific color can be selected by means of optical diffraction, and the wrinkle-like pattern period can also be controlled by altering the conditions of the dip-coating process.

While most optical applications of M13 bacteriophage have been based on diffraction, the ordered structure also possesses great potential for bio-lasers. Due to the coating protein of the M13 bacteriophage, fluorescent dyes can be attached to the M13 bacteriophage, becoming immobilized [16,17]. This is reminiscent of a distributed feedback laser (DFB), where the corrugated structure embedded in an active gain medium gives rise to a constructive interference of the propagation light under Bragg diffraction conditions [18,19].

In this work, we investigate the modal gain of an Alexa fluor 488/M13 bacteriophage hybrid structure, where the dye (Alexa fluor) is coated on the wrinkle-like ordered structure of the M13 bacteriophage. To obtain optical modal gain, we used the variable stripe length method (VSLM), where amplified spontaneous emission (ASE) was measured with increasing stripe length. Regarding the waveguide effect, the wrinkle assembly lines were aligned to be either parallel or perpendicular to an excited optical stripe. We found that the edge emission of ASE showed a strong polarization anisotropy [20,21,22,23], and the modal gain spectrum also showed a spectral etalon due to the corrugated structure. These results make it possible to obtain a degree of linear polarization (DOLP) and an effective cavity size. 

## 2. Materials and Methods

M13 bacteriophage was synthesized according to the known protocol [14], and dissolved in distilled water to prepare a precursor solution. As shown schematically in Figure 1a, the dip-coating method was used to fabricate thin M13 bacteriophage film [13]. While the solvent evaporates, a self-assembly is induced through three states, immersion, withdrawal, and evaporation [24]. For the immersion stage, a glass substrate was immersed into the M13 bacteriophage solution for a certain dwell time to result in complete wetting of the substrate. During the withdrawal stage, the substrate is pulled up vertically from the solution at a constant withdrawal velocity, which is controlled automatically by syringe pump, Legato 180 (KD Scientific, Holliston, MA, USA). As the substrate is pulled up, a thin layer of the precursor solution is dragged with it, but any excess liquid drains into the solution reservoir. As a result, the surface becomes meniscus-like. However, water solvent evaporates from the layer, the combined effect of capillary flow, concentration gradient along the meniscus, and substrate withdrawal velocity give rise to a periodic structure for the M13 bacteriophage [13,14,15]. For dye-deposition on the ordered structure of M13 bacteriophage, drops of streptavidin conjugate Alexa-fluor 488 solution (Thermo Fisher Scientific, Waltham, MA, USA) are casted. Because the streptavidin leads to an effective binding to M13 bacteriophage, the dye (Alexa-fluor 488) also takes on an ordered periodic structure, as shown in Figure 1a.

In Figure 1b, the ordered assembly of the M13 bacteriophage can be seen on the basis of the atomic force microscopy (AFM) image, where the corrugated structure shows an aperiodic regularity. Figure 1c shows the schematics of polarization-resolved edge emission setup. For excitation, a 404 nm picosecond pulsed laser (PicoQuant, Berlin, Germany) is used with a 40 MHz repetition rate. The linear polarization of excitation is aligned to be either parallel or perpendicular to the wrinkle line of the M13 bacteriophage, denoted by horizontal (H) and vertical (V) polarization states, respectively. The excitation beam becomes an optical stripe on the sample by using a cylindrical lens, and we selected a uniform range (<500 μm) regarding the intensity distribution along the stripe. The length of the optical stripe is controlled by moving a beam block. With respect to optical diffraction effects, a thin black blade was used. As the optical stripe length increases, the edge emission of the amplified spontaneous emission (ASE) exhibits exponential growth. Therefore, optical modal gain can be obtained by VSLM [25]. We also analyzed the polarization of the edge emission. All measurements were performed at room temperature with the same excitation power of 1 mW. 

## 3. Results and Discussion

In Figure 2, the polarization dependence of the ASE spectrum is presented for H- and V-polarized excitation, respectively. For V-polarized excitation, the ASE spectrum became enhanced significantly in long wavelengths (Figure 2d) compared to the ASE spectrum with H-polarized excitation (Figure 2a). Regarding the corrugated structure of the dye-M13 bacteriophage, this result is possibly associated with two effects, one-dimensional interference grating and polarization dependence in the waveguide. 

As shown in Figure 1b, the wrinkles have a period a few μm in height, but the variations in height occurs at an order-of-magnitude smaller scale (~0.1 μm). Therefore, this could be a case of one-dimensional interference Bragg grating in DFB lasers, where the periodic changes in refractive index cause reflection back into the cavity. When the ASE propagates along the perpendicular direction of the wrinkles (V-polarized excitation), the corrugated structure seems to result in a constructive interference at long wavelengths of ASE (>560 nm) despite the aperiodic irregularity and the inhomogeneity of the wrinkles. On the other hand, light propagation along the wrinkles seems to be interrupted in the H-polarized excitation configuration. It can be seen in the AFM image in Figure 1b that the wrinkle lines are not straight, but rather sinusoidal in the lateral plane with a period of several μm. In this case, the effective cavity length is expected to be large compared to that of the V-polarized excitation. Hence, no interference grating effect occurs.

With respect to the polarization-dependent internal reflection in the waveguide, the edge emissions under both the H- and V-excitation are expected to have a polarization anisotropy. For example, the H-polarized excitation configuration results in light amplification along the wrinkle lines. In this case, the transverse magnetic (TM)-mode, which corresponds to the analyzer angle of *θ* = 90°, seems to be suppressed compared to the transverse electric (TE)-mode (*θ* = 0°). On the other hand, the corrugated structure may give rise to a constructive interference with a relatively large degree of polarization under the configuration of V-polarized excitation. Nevertheless, this is difficult to predict due to the aperiodicity and inhomogeneity. 

In Figure 2b,e, the polarization dependence of the ASE spectrum for H- and V-polarized excitation are compared. With H-polarized excitation, we found that TE-mode (*θ* = 0°) was enhanced at all wavelengths of the ASE spectrum. On the other hand, with V-polarized excitation, the opposite result was obtained, i.e., the ASE intensity of the TM-mode (*θ* = 90°) became larger than that of the TE-mode (*θ* = 0°). In Figure 2c,f, polarization anisotropy is evaluated in terms of linear degree of polarization (LDOP), which is defined as
(1)$$\mathrm{LDOP}\;=\frac{I_{\theta =0{}^{\circ}}-I_{\theta =90{}^{\circ}}}{I_{\theta =0{}^{\circ}}+I_{\theta =90{}^{\circ}}},$$
where $I_{\theta =0{}^{\circ}}$ and $I_{\theta =90{}^{\circ}}$ are ASE intensity at analyzer angles of θ=0° and θ=90°, respectively.

In the case of H-polarized excitation, ASE is emitted along the wrinkle lines. According to the Fresnel equations, the TE-mode (*θ* = 0°) ASE is confined to the wrinkle waveguide with large reflection coefficients, but the TM-mode (*θ* = 90°) ASE transmits beyond the wrinkle barrier. When the sample is rotated for V-polarized excitation, the TM-mode (*θ* = 90°) ASE can be observed to be enhanced. In other words, the TE-mode (*θ* = 0°) is suppressed. 

Figure 3a,d show the stripe length dependence on ASE intensity for H- and V-polarized excitation, respectively. When the stripe is considered to be a one-dimensional waveguide, the edge emission intensity increases exponentially with increasing stripe length (x). The stripe length dependence of ASE ($I_{\mathrm{ASE}}\left(\hbar \omega ,x\right)$) at a selected emission energy (ℏω) is given by [26]
(2)$$\frac{dI_{\mathrm{ASE}}\left(\hbar \omega ,x\right)}{dx}=g\left(\hbar \omega ,x\right)\cdot I_{\mathrm{ASE}}\left(\hbar \omega ,x\right)+J_{\mathrm{spon}}\Omega$$
where g, $J_{\mathrm{spon}}$, and Ω are the optical gain coefficient, spontaneous emission density, and solid angle of the edge emission, respectively. Assuming that g is independent of the stripe length, $I_{\mathrm{ASE}}\left(\hbar \omega ,x\right)$ can be obtained as [26]
(3)$$I_{\mathrm{ASE}}\left(\hbar \omega ,x\right)=\frac{J_{\mathrm{spon}}\Omega }{g\left(\hbar \omega \right)}\left(e^{g\left(\hbar \omega \right)x}-1\right).$$

However, this function is no longer valid when gain saturation becomes significant. With increasing stripe length, the ASE intensity deviates from Equation (3), as shown in Figure 3a,d. In this case, gain needs to be analyzed in terms of stripe length dependence. It is noticeable that gain saturation is associated with stripe length dependence, although the terminology of gain saturation is often used to describe excitation dependence. Specifically, stripe length dependence is a consequence of light-matter interaction along the stripe. With increasing ASE intensity along the stripe, carrier consumption also increases. Therefore, carrier diffusion occurs, and an inhomogeneous carrier distribution is also induced. As a result, gain saturation becomes dependent on stripe length with increasing stripe length. 

Regarding the fundamental differential equation (Equation (2)), the gain coefficient can be obtained experimentally as follows [27]:(4)$$g\left(\hbar \omega ,x\right)=\frac{\frac{dI_{\mathrm{ASE}}\left(\hbar \omega ,x\right)}{dx}-J_{\mathrm{spon}}\Omega }{I_{\mathrm{ASE}}\left(\hbar \omega ,x\right)}$$
where $J_{\mathrm{spon}}\Omega$ is obtained from $\frac{dI_{\mathrm{ASE}}\left(x\right)}{dx}$ at x=0. Figure 3b,e show the stripe length dependence of the gain spectrum for H- and V-polarized excitation, respectively (Appendix A). Regarding the validity of the one-dimensional model, gain coefficients were plotted from a stripe length of 100 μm, which is comparable to the stripe width. It is difficult to find significant differences in the stripe length dependence of gain between the H- and V-polarized excitations. 

However, the stripe-length-dependent gain spectrum of H (Figure 3c)- and V (Figure 3f)-polarized excitation display different spectral modulations (Appendix A). The gain spectrum of V-polarized excitation is enhanced at long wavelengths compared to that of H-polarized excitation. Interestingly, both gain spectra show an etalon-like periodic modulation, but the wavelength periods (Δλ) are different. The wavelength period ($\Delta \lambda_{V}$) in the gain spectrum for V-polarized excitation is longer than that ($\Delta \lambda_{H}$) in the gain spectrum for H-polarized excitation.

Theoretically, the wavelength periods of ($\Delta \lambda_{H}$) and ($\Delta \lambda_{V}$) can be associated with a mode in an optically confined cavity. Given a dominant wavelength ($\lambda_{0}$), the mode spacing (Δλ) is given by [28]
(5)$$\Delta \lambda =\frac{\lambda_{0}^{2}}{2L\left(n-\lambda_{0}dn/d\lambda \right)}$$
where n is the refractive index at $\lambda_{0}$, and L is the cavity length. Alexa-fluor-488 solution was used to measure the refractive index using the interference fringe change via refraction [29]. 

Figure 4a,b show the stripe length dependence of L and Δλ for H- and V-polarized excitation, respectively. For example, given the gain spectrum with H-polarized excitation and a 100 μm stripe length, the wavelength period ($\Delta \lambda_{H}=16$ nm) was obtained from a cavity length of L=1.6 μm (Appendix A). In the case of H-polarized excitation, it is probable that the wrinkles aligned parallel to the amplifying direction of the edge emission provide localized micro-cavities. Although the exact mechanism is unknown, this result suggests that the winkle-like structure exhibits local deviations from an ideal straight waveguide, according to the lateral AFM image [15]. Interestingly, the size of the effective cavities exhibits a period with increasing stripe length. This could be associated with the local structure being similar to that of the random lasing mechanism of emitting scatters. On the other hand, the wavelength period ($\Delta \lambda_{V}∼44$ nm) of V-polarized excitation was observed to be long compared to that of H-polarized excitation, and the corresponding cavity size was in the sub-micron range. It is noticeable that the wrinkle alignment direction is perpendicular to the amplifying direction. Therefore, the degree of optical mode confinement decreased. Although the average distance between the wrinkle crests is a few microns, the morphology changes gradually within the period. This gives rise to localized cavities smaller than the wrinkle period. Nevertheless, it is advantageous that the amplified edge emission becomes polarized with the periodic structure.

It is also interesting that the etalon is barely observable in the ASE spectrum. This can be attributed to the aperiodic irregularity of the corrugated structure, whereby mode selection becomes inefficient. Nevertheless, the gain etalon suggests that amplification along the corrugated structure is very sensitive to the effective size. Therefore, the gain etalon is useful for determining the optimum cavity length of the aperiodic corrugated DFB nano-bio-laser. 

## 4. Conclusions

A periodic dye-virus hybrid structure (Alexa-fluor 488/M13 bacteriophage) was studied in terms of the polarization dependence of ASE. We observed that the edge emission showed an etalon-like periodic modulation in the ASE spectrum with a strong polarization anisotropy, and the period contributed to a localized cavity mode within the wrinkle-like structure. We also found that modulation affected the gain spectrum, and was dependent on the excitation polarization direction with respect to the wrinkle alignment.

## Figures and Tables

**Figure 1 nanomaterials-11-03309-f001:**
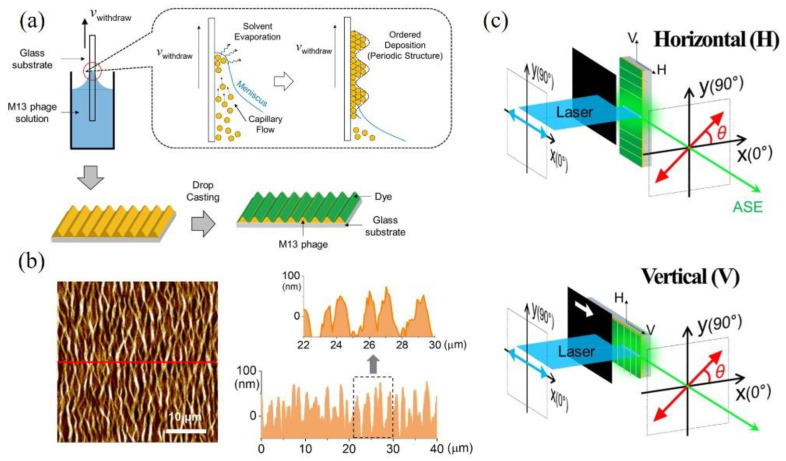
(**a**) After the periodic structure of M13 bacteriophage is formed by the dip-coating method, where evaporation induces self-assembly, dye (Alexa-Flour488) is deposited on the ordered structure of M13 bacteriophage thin film. (**b**) AFM image showing the corrugated structure of M13 bacteriophage. (**c**) Polarized edge emission is measured to obtain the stripe length dependence of amplified spontaneous emission (ASE), where the optical stripe generated by a 400 nm laser is either horizontal (H) or vertical (V) to the corrugated direction of M13 bacteriophage.

**Figure 2 nanomaterials-11-03309-f002:**
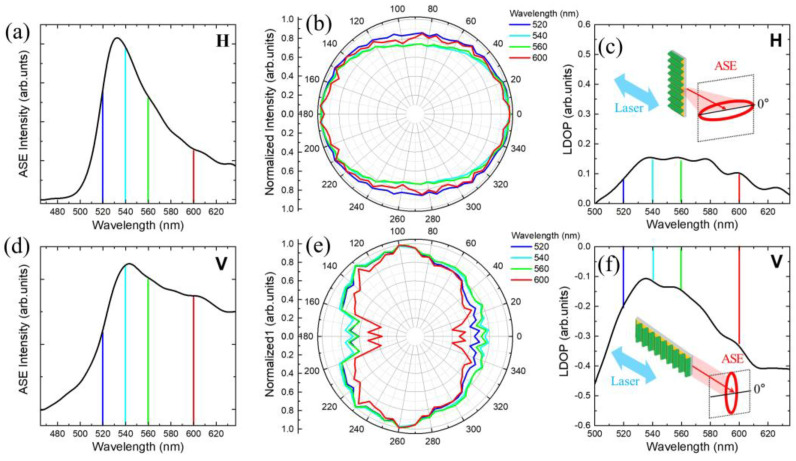
(**a**,**b**) Horizontally and (**d**,**e**) vertically polarized excitation; the ASE spectrum and analyzer angle dependence are shown for selected wavelengths of blue (520 nm), cyan (540 nm), green (560 nm), and red (600 nm), respectively. (**c**,**f**) Linear degree of polarization (LDOP) was estimated for horizontally (H) and vertically (V) polarized excitation, respectively.

**Figure 3 nanomaterials-11-03309-f003:**
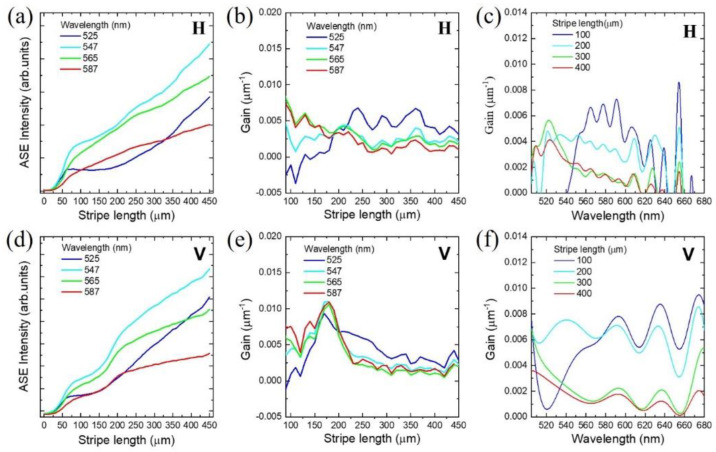
For selected wavelengths of blue (525 nm), cyan (547 nm), green (565 nm), and red (587 nm), stripe length dependence of (**a**,**d**) ASE intensity and (**b**,**e**) gain were obtained for horizontally (H) and vertically (V) polarized excitation, respectively. (**c**,**f**) Gain spectrum obtained with H- and V-polarized excitation show a significant difference.

**Figure 4 nanomaterials-11-03309-f004:**
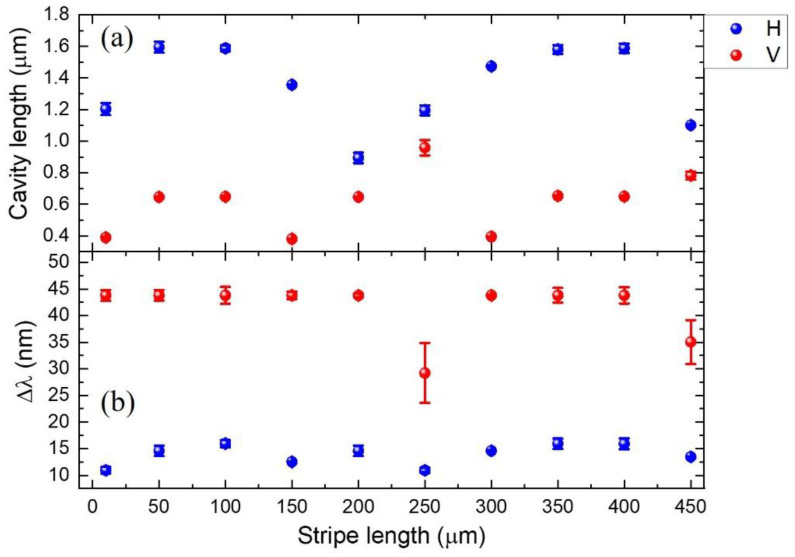
(**a**) Cavity length and (**b**) mode spacing under H- and V-polarized excitation were plotted for increasing stripe length.

## Data Availability

The data presented in this study are available on request from the corresponding author.

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
