# Peer review of "Polarization Angle Dependence of Optical Gain in a Hybrid Structure of Alexa-Flour 488/M13 Bacteriophage"

_nanomaterials, 2021, doi:10.3390/nano11123309_

Round 1

Reviewer 1 Report

This manuscript reports an investigation on the polarization angle dependence of ASE in a hybrid structure.

  1. At Page3, Line103-104: “for V-polarization excitation, the ASE spectrum becomes enhanced significantly in long wavelengths”, the explanation of this phenomenon is unclear, why does the periodicity of bacteriophage lead this situation? Maybe the author can provide the more detail explanation.

  1. Figure4 shows the cavity length and mode spacing under the various stripe lengths. How to obtain these results? Simulated gain spectrum?

Why cannot observe this etalon effect at experiments spectrum (for example in Figs. 2a and 2d)?

  1. Page5, Line177: the wavelength period of V-polarization excitation should be 40nm?

  1. In addition to the ASE, there are many works have been proposed to investigate the optical polarization properties of biomolecule through the laser emission: [Ref: ACS Photonics 7.8 (2020): 1908-1914; Physical Review B 92.21 (2015): 214202; ACS Nano 2021, 15, 5, 8965–8975; Physical Review A 86.4 (2012): 043817.] I would recommend a brief introduction of molecular polarization in laser emission to strengthen the broad interest. 

Reviewer 2 Report

The authors discussed the polarization dependence of the gain model in a period hybrid structure by using the variable strength length method (VSLM). This method has been used by the authors in analyzing optical gain in other nanostructures. The work in the manuscript is interesting and maybe published in nanomaterials in a major revision.

  1. the authors discussed the optical gain of a period hybrid structure and a micro-scale bio-laser can be realized if enough gain and strong polarization confinement is provided. The author should discuss the potential or limitations for a laser generation in their structure. Besides, the hybrid structure is similar to the structure of a DFB laser, which can be easily realized. Thus, the authors should compare their structure with a DFB laser.
  2. the authors should provide an image (physical map) of the hybrid structure assembled in the experiment; some parameters of the hybrid structure has to be provided, such as the the period of the order structures, the shape of the structure.
  3. the reason causing the difference in polarization degrees of the H-polarized excitation and V-polarized excitation has to be discussed fully.
  4. what is the energy /power density of the excitation beams? In the discussion of Eqs. (3) and (4), the authors said the gain saturation become significant, the author should explained that in which conditions the gain saturation become important.
  5. The first paragraph in the Results and Discussion Section should be introduced in the introduction section; the first paragraph is a background introduction, not results of the author’s work.
